# Polyaniline-Coated Porous Vanadium Nitride Microrods for Enhanced Performance of a Lithium–Sulfur Battery

**DOI:** 10.3390/molecules28041823

**Published:** 2023-02-15

**Authors:** Jingjie Lv, Haibo Ren, Ziyan Cheng, Sang Woo Joo, Jiarui Huang

**Affiliations:** 1Key Laboratory of Functional Molecular Solids of the Ministry of Education, Anhui Laboratory of Molecule-Based Materials, College of Chemistry and Materials Science, Anhui Normal University, Wuhu 241002, China; 2School of Materials Science and Engineering, Modern Technology Center, Anhui Polytechnic University, Wuhu 241000, China; 3School of Mechanical Engineering, Yeungnam University, Gyeongsan 712749, Republic of Korea

**Keywords:** VN, microrods, sulfur host, cathode, lithium–sulfur battery

## Abstract

To solve the slow kinetics of polysulfide conversion reaction in Li-S battery, many transition metal nitrides were developed for sulfur hosts. Herein, novel polyaniline-coated porous vanadium nitride (VN) microrods were synthesized via a calcination, washing and polyaniline-coating process, which served as sulfur host for Li-S battery exhibited high electrochemical performance. The porous VN microrods with high specific surface area provided enough interspace to overcome the volume change of the cathode. The outer layer of polyaniline as a conductive shell enhanced the cathode conductivity, effectively blocked the shuttle effect of polysulfides, thus improving the cycling capacity of Li-S battery. The cathode exhibited an initial capacity of 1007 mAh g^−1^ at 0.5 A g^−1^, and the reversible capacity remained at 735 mAh g^−1^ over 150 cycles.

## 1. Introduction

The lithium–sulfur battery is rapidly developing as a promising rechargeable battery because of its high energy density, low cost, nontoxicity, and abundance of raw materials [1,2]. However, due to the shuttle effect of soluble polysulfides, poor conductivity and volume change of sulfur during the discharge procedure, and corrosion and dendrites of the lithium anode, the low utilization rate of sulfur and inferior cycling stability have limited the battery life, the application of the Li-S battery is hindered [3,4,5]. Therefore, it is very important to find suitable catalysts to prevent the shuttle effect of polysulfides, promote redox kinetics, and enhance cycling capacity of the Li-S battery.

To anchor lithium polysulfides (LiPSs), many sulfur hosts such as metal oxides/suldes/nitrides have been developed in recent years [6,7]. Among these sulfur hosts, polar materials including vanadium nitride (VN) [8], titanium nitride (TiN) [9], cobalt nitride (Co_4_N) [10], and other transition metal nitrides are excellent sulfur hosts because they can inhibit the diffusion of polysulfides through forming S-M-N bonds, improving the cycling stability [11,12]. In addition, the d-layer electron orbitals of the metal atoms in the transition metal nitride lattice overlap each other, exhibiting conductivity similar to that of metals [13,14]. As a transition metal nitride, VN can trap intermediates and limit the dissolution of polysulfides in the electrolytes [15,16,17], and the porous rod-like microstructure has a large specific surface area, which possesses a lot of cavities to overcome the volume expansion during cycling [18].

In recent years, many polymers and carbon-based materials have become hot spots for lithium–sulfur batteries [19,20,21]. For example, various S-doped carbon nanomaterials were developed and served as hosts for sulfur cathode in order to achieve rapid polysulfide redox reaction [22,23]. Traditional conducting polymers are also well known for their good flexibility and good conductivity [24,25]. Furthermore, N-containing groups and conjugated structures in p-conducting polymer can be used as active sites to anchor polysulfides [26]. As a result, these conductive polymers are commonly used as coatings, conducting agents, diaphragm modifiers, and binders for lithium–sulfur battery. For instance, poly(allylamine hydrochloride) and poly(styrene sulfonate sodium salt) were alternately coated on the sulfur particles, and then a polyaniline (PANI) layer was in situ polymerized on the sulfur shell, which exhibited a capacity of 641 mAh g^−1^ over 300 cycles at 1 C [27]. A group prepared hollow PANI/sulfur/PANI composite via sequentially depositing method using silicon spheres as templates [28]. In this structure, S particles were wrapped by the PANI layers, and the PANI layer facilitates ion penetration and inhibits polysulfide diffusion. Thus, the PANI/sulfur/PANI composite delivered a capacity of 572.2 mAh g^−1^ over 200 cycles at 0.1 C. Therefore, PANI is an up-and-coming cathode material because of the high conductivity and good electrochemical activity [29].

In this study, we synthesized novel polyaniline-coated VN porous microrods via a calcination, washing and polyaniline-coating process. When served as a sulfur host for Li-S battery, the polyaniline-coated VN porous microrods exhibited high electrochemical performance. The porous structure of VN microrods with a high specific surface area provided abundant active sites for electrochemical reaction, enhanced the transfer of electrons and ions, and supplied large interspace to overcome the volume expansion of cathode. The outer layer of polyaniline further enhanced the cathode conductivity, and nitrogen-containing groups in the polymer also prevented the polysulfides from shuttling.

## 2. Results and Discussion

### 2.1. Characterization

The fabrication process of porous VN/S@PANI microrods is shown in Figure 1. First, porous VN microrods were synthesized by one-step calcination at a high temperature of 700 °C in tube furnace in an Ar atmosphere. The porous VN/S microrods were synthesized via a sulfur melt-diffusion process. To promote cycling stability of the cathode, VN/S microrods were wrapped by PANI layer via an in situ polymerization of aniline.

Figure 2 displays the SEM and TEM images of VN microrods, VN/S microrods and VN/S@PANI microrods. From Figure 2a, the products were mainly composed of porous rod-like VN with a diameter of ~600 nm and a length of ~15 μm. The high-magnification SEM image and TEM image of the VN microrods (Figure 2b,c) verify their porous structure. Figure 2d,f show the SEM and TEM images of the VN/S microrods. Clearly, the VN/S sample still retained the initial rod-like morphology. However, the surface of the microrods became smooth and the abundant pores could not be observed due to the sulfur filling. The SEM images and TEM image of the VN/S@PANI microrods are shown in Figure 2g–i. Compared with VN/S microrods, the surface of PANI coated microrods is smoother, indicating that the PANI layer has wrapped on the VN/S microrods.

The phase composition of these samples was investigated by XRD. In Figure 3a, the peaks of the porous VN microrods belong to cubic vanadium nitride (JCPDS No. 35-0768), and there are no other peaks, indicating that the VN product was pure. In addition, the diffracted peaks of VN/S composites contain the peaks of VN and sulfur powder (JCPDS No. 08-0247), indicating that VN was successfully combined with sulfur. Furthermore, the diffraction peaks of the VN/S@PANI microrods are similar to that of the VN/S microrods. Figure 3b exhibits the Raman spectrum of pure porous VN microrods, VN/S microrods and VN/S@PANI microrods. In the Raman spectrum of the VN microrods, characteristic peaks at 137, 271, 402, 519, 689 and 988 cm^−1^ were attributed to cubic VN [30]. Compared with the pure VN microrods, the VN/S microrods presented two additional peaks at 215, 470 cm^−1^, corresponding to the sulfur, and suggesting successful recombination of sulfur with the VN microrods [31]. For the Raman spectrum of the VN/S@PANI microrods, a peak at 1380 cm^−1^ is ascribed to the polaron C-N^+^ vibration, whereas a peak at 1569 cm^−1^ is due to the tensile vibration of C=C in the quinine ring [32]. The FTIR spectra of VN microrods, VN/S microrods and VN/S@PANI microrods are exhibited in Figure 3c. For the FTIR spectrum of VN/S@PANI microrods, two peaks at 1577, 1499 cm^−1^ correspond to the basic vibration of the quinine ring and benzene ring, respectively, and the C-N in-plane vibrations are at 1297 and 1238 cm^−1^. In addition, a peak at 1129 cm^−1^ is ascribed to the C=N stretching vibration [33]. The peaks of VN/S@PANI microrods indicate that aniline monomer has been polymerized on the surface of the VN/S microrods. Moreover, the EDS spectrum of VN/S@PANI microrods (Figure 3d) indicates the presence of V, N, C, O, and S elements. The elemental scan line (Appendix A) shows the proportion of the elements on the surface.

A high-resolution TEM image displayed in Figure 3e indicates its highly crystalline nature. A lattice fringe of 0.24 nm matches the (111) plane of cubic VN. Figure 3f shows the SAED pattern of VN microrods, which can be ascribed to (111), (200), (220) and (420) planes of VN [33]. The elemental mapping images of VN/S@PANI microrods are displayed in Figure 4, indicating that V, N, S, C, and O elements are distributed evenly in VN/S@PANI composite.

Figure 5 shows the N_2_ adsorption/desorption isotherms of VN microrods, VN/S microrods and VN/S@PANI composites. In Figure 5a, the specific surface area of porous VN microrods was up to 310.6 m^2^ g^−1^, and the pore size was mostly distributed in 0.1–2.5 nm. Its ultra-high specific surface area with high porosity is favorable for loading more sulfur and provides more active sites for sulfur oxidation and reduction [34]. The specific surface areas of VN/S and VN/S@PANI decreased to 4.0 and 6.3 m^2^ g^−1^, respectively, demonstrating that sulfur species were composited with these sulfur hosts. In Figure 5e,f, the disappearance of mesopores indicates that sulfur was impregnated in the mesopores of VN microrods. Appendix A shows the TGA curves of VN/S microrods and VN/S@PANI composite. For the VN/S microrods, the total loss between 100 °C–280 °C is 69.9 wt%, which is attributed to sulfur evaporation. For the VN/S@PANI composite, the total loss between 100 °C–280 °C was 81.2 wt%, which was attributed to S volatilization and decomposition of PANI. The total loss between 280 °C–550 °C was 4.3 wt%, which was attributed to the decomposition of PANI.

XPS spectra of the VN/S@PANI composite are exhibited in Figure 6. The XPS survey spectrum verifies the presence of V, N, C, O and S elements in the sample. Figure 6b displays the existence of V-N (513.8 eV, 521.5 eV), V-N-O (515.0 eV, 522.8 eV) and V-O (516.9 eV, 524.4 eV) bonds. Figure 6c exhibits the peaks of N 1s at 397.5, 398.5, 399.5, and 400.9 eV, which are assigned to N-V, -N=, -NH-, and -NH_2_^+^-, respectively [35]. Figure 6d exhibits four peaks of C 1s at 284.5, 285.3, 286.4, and 288.4 eV, corresponding to the C-C, C-N, C-O, and C=O, respectively [15]. Figure 6e exhibits three peaks of O 1s at 529.9, 530.9 and 532.1 eV, which are assigned to the V-O, C-O and O-H, respectively [36]. In Figure 6f, the S 2p spectrum exhibited four peaks at 164.0, 165.0, 168.4, and 169.6 eV, which are ascribed to S-C, sulfate, S 2p_1/2_, and S 2p_3/2_, respectively. The sulfates are attributed to sulfur oxidation in the air [37].

### 2.2. Electrochemical Performance

The specific capacity of the cathode in this work is the mass specific capacity, that is, the quantity of electricity can be discharged per unit mass of the cathode material. It can be expressed as the following formula. *S_c_* = *I_d_* × *t*, where *S_c_* is the specific capacity of the cathode, *I_d_* is the current intensity per unit mass, which is also called current density, and *t* is the charge and discharge time of the battery. In Figure 7a, the VN/S@PANI cathode delivered a high initial capacity of 1007 mAh g^−1^ at 0.5 A g^−1^, and it maintained at 735 mAh g^−1^ over 150 cycles, which is higher than that of pristine VN/S microrod cathode (372 mAh g^−1^). The charge/discharge curves are displayed in Figure 7b. As for the cycling performance displayed in Figure 7c, it was maintained at 458 mAh g^−1^ over 400 cycles, which was also higher than that (254 mAh g^−1^) of VN/S microrod cathode. It can be seen that the polyaniline coating significantly improved its cycling stability. As shown in Figure 7d, the rate capacities were 1269, 947, 803, 710, and 561 mAh g^−1^ at the current densities of 0.1, 0.2, 0.5, 1, and 2 A g^−1^, respectively. However, the capacity of VN/S cathode was only 87 mAh g^−1^ at 2 A g^−1^. Appendix A lists the cyclic performance of some sulfur hosts for Li-S battery. The cyclic performance of VN/S@PANI was superior to those of the VN/S microflowers [11], PANI@BDC/S [35], S/VN@CNFs [38], VN-NCNFs/S nanofibers [39], V_2_O_3_-VN@NC/S [40], and some other vanadium-based composites [41,42,43,44]. In Figure 7e, VN/S@PANI cathode delivered 338.6 mAh g^−1^ after 500 cycles at 50 °C, and the Coulombic efficiency remained at 99.9%. Local high temperature can cause uneven electrochemical reaction in the whole battery, and the dissolution and shuttling of polysulfide in the heat-increasing region become even more obvious [45]. The good high temperature resistance is due to the unique microstructure which can effectively inhibit polysulfide shuttling.

Figure 8a exhibits the initial three CV curves of VN/S@PANI cathode. On the first curves, reductive peaks at 2.33 and 2.0 V, and one oxidative peak at 2.35 V were observed. In later cycles, the reduction peaks of 2.34 and 2.05 V were due to the formation of Li_2_S_4_ and Li_2_S_2_/Li_2_S, respectively. The oxidation peak of 2.36 V is related to the conversion of Li_2_S to Li_2_S_8_ and sulfur. Similarly, the VN/S microrod cathode exhibited two reduction peaks at 2.30 and 2.05 V, and one oxidation peak at 2.39 V on first curves (Appendix A). On later curves, the cathode displayed two reduction peaks at 2.30 and 2.06 V and one oxidation peak at 2.36 V. The shift of redox peak is due to the rearrangement of active sulfur. Small peak shifts of oxidation and reduction can be observed in VN/S@PANI, proving admirable reversibility of VN/S@PANI cathodes. Additionally, VN/S@PANI cathode have two close peaks at 2.31 V, 2.36 V from subsequent cycles, corresponding to the oxidation of short-chain Li_2_S/Li_2_S_2_ to long-chain Li_2_S_x_ and finally to S_8_, respectively. Only a broad oxidation peak was found for VN/S cathodes because of the low redox kinetics of LiPSs and severe polarity [46].

The electrochemical kinetics were further investigated by the CV test between 0.1 and 1.0 mV s^−1^, as exhibited in Figure 8b. The CV curves at different rates were skewed due to the polarization, and the potential drift is mainly caused by the depolarization-calibrated ohmic resistors [47]. Li^+^ diffusion rate could be calculated by following formula:*I_p_* = 2.69 × 10^5^*n*^1.5^*SD_Li +_*
^0.5^C*_Li +_*ν^0.5^(1)
where *Ip*, *n*, *S*, and *v* are peak current, electron transfer number, cathode area, Li^+^ transfer rate, Li^+^ concentration, and sweep rate, respectively. In Li-S batteries, *Ip*, *n*, *S*, values are constant [48]. The slope of the slash corresponds to that of the electrochemical process. During each electrochemical reaction, the slope value of the VN/S@PANI cathode (Figure 8c) was higher than that of the VN/S cathode (Appendix A), indicating a faster Li^+^ diffusion rate and outstanding conversion redox kinetics [38]. It can be ascribed to efficient adsorption of polysulfides by conductivity polyaniline which promotes Li^+^ diffusion and accelerates the conversion kinetics of LiPSs. Figure 8d is derived from *i* = *k*_1_*ν* + *k*_2_*ν*^1/2^, where *k*_1_*ν* and *k*_2_*ν*^1/2^ are related to the contributions of capacitance and diffusion, respectively. The results show that the ratio of capacitance contribution increased with scanning speed. The capacitance control ratio of VN/S@PANI is obviously higher than that of the VN/S cathode (Appendix A).

Galvanostatic intermittent titration technique (GITT) test was carried out when the battery relaxed for 2 h at 0.2 A g^−1^. Ten cycles later, the Li-S battery was charged and discharged for 300 s, as shown in Figure 9a,b. Obviously, compared with the VN/S microrod (Figure 9b), the VN/S@PANI cathode (Figure 9a) displayed smaller discharge/charge polarization voltage plateaus (ΔE) [49]. The internal resistance at different discharge/charge procedures could be calculated by following equation [50]:
ΔR_internal_ (Ω/g) = ΔU_overpotential_/I_appiled_ m(2)

The ΔU_overpotential_ differs greatly from the quasi-open circuit voltage and closed-circuit voltage. The I_applied_ is the applied current, m is the mass of active material of cathode. The reaction impedances of VN/S and VN/S@PANI cathodes are shown in Figure 9c,d. Compared with the VN/S, the VN/S@PANI has higher conductivity, higher electron and ion transfer efficiency, smaller energy barrier and smaller interfacial resistance, thus enhancing electrochemical performance [51].

The absorption of VN microrods and VN@PANI composites towards Li_2_S_6_ is displayed in Figure 10a. By observing absorption in real time, Li_2_S_6_ solution containing VN turned to pale yellow after absorbing for 24 h, while the Li_2_S_6_ solution containing VN@PANI became colorless, indicating that the adsorption capacity of the VN@PANI for Li_2_S_6_ was significantly stronger than that of the porous VN microrods. In addition, to further prove the adsorption capability of the VN microrods and VN@PANI microrods for polysulfide. After adsorption for 24 h, the solution was tested by UV-vis spectroscopy (Figure 10b), and two peaks at 420 nm and 460 nm correspond to S_4_^2−^ and S_6_^2−^, respectively [52]. Among these Li_2_S_6_ solutions, the peak intensity of Li_2_S_6_ solution containing VN@PANI was the weakest, indicating that the absorbability of the VN@PANI microrods to polysulfide was better than that of the porous VN microrods.

Figure 11 shows the electrochemical impedance spectroscopy (EIS) of VN/S@PANI and VN/S cathodes before and after 500 cycles. In Figure 11a, a concave semicircle represented charge transfer resistance between the cathode and the electrolyte, corresponding to the *Rct* value in the equivalent circuit [53]. The inclined line corresponds to Li^+^ diffusion in the electrolyte, which is described as Warburg impedance (W_o_) [53]. The insets in Figure 11 show the equivalent circuit diagram before and after the 500 cycles. In the equivalent circuit diagram, *R_1_* represents the internal resistance of the electrolyte, *R_2_* represents the internal resistance of the solid electrolyte interface (SEI) film correlated with insoluble Li_2_S_2_/Li_2_S, *Rct* is related to the charge transfer resistance and the electrode chemical kinetics. *CPE1* represents the capacitance of the electrode body in the high frequency region, and *CPE2* represents the capacitance of the charge transfer process at the sulfur–electrolyte interface, *Wo* is the semi-infinite Warburg diffusion impedance of long chain LiPSs [49,53]. The charge transfer resistance of the VN/S@PANI cathode (45.1 Ω) was obvious smaller than that of VN/S cathode (136.8 Ω), and the kinetic redox property of the polysulfide conversion was improved due to the increase in conductivity via polyaniline coating. In Figure 11b, the VN/S and VN/S@PANI cathodes after 500 cycles have an additional concave semicircle in the high-frequency region due to the formation of SEI film (R_SEI_). The *Rct* of VN/S@PANI cathode (76.5 Ω) was lower than that of the VN/S cathode (133.5 Ω) after the cycling test, which may be related to electrolyte permeation and redistribution of polysulfide. The reduced *Rct* was mainly ascribed to the unique microstructure in which insulating sulfur is well encapsulated within the polyaniline layer, thereby accelerating the transfer of electrons and ions. The SEM and TEM images of VN/S@PANI cathode after 500 cycles are exhibited in Appendix A. The results showed that the VN/S@PANI composite still maintained the rod-like morphology during the cycling process, which indicated that the VN/S@PANI composite had excellent stability to enhance the cycling performance of the cathode.

## 3. Materials and Methods

### 3.1. Preparation of Porous VN Microrods

NaVO_3_ (0.2 g) and C_3_H_6_N_6_ (0.6 g) were mixed and transferred into a tube furnace in N_2_ atmosphere and calcinated at 700 °C for 2 h. Finally, the black product was washed and dried at 90 °C for 10 h.

### 3.2. Preparation of Porous VN/S Microrods

Typically, 0.2 g porous VN microrods and 0.5 g sulfur were mixed evenly in a container filled with Ar. After that, the vessel was heated at 155 °C for 24 h.

### 3.3. Preparation of VN/S@PANI Microrods

Concentrated sulfuric acid (2 mL) was added into 98 mL deionized water. Then, 0.1 g VN/S microrods and 20 μL of aniline were added into the above dilute sulfuric acid solution. After that, 0.5705 g ammonium persulfate was put in the solution. The mixture was stirred in an ice water bath at 0 °C for 12 h, and then washed and dried at 60 °C for 8 h.

### 3.4. Study of LiPSs Adsorption Experiment

The Li_2_S_6_ solution was compounded through adding S and Li_2_S (5:1 molar ratio) into the DME (dioxolane) Dol (1,2-dimethoxyethane) solvent (1:1) in argon atmosphere. The mixtures were kept at 80 °C for 20 h. After that, 8 mg porous VN microrods and porous VN@PANI microrods were added into 4 mL of Li_2_S_6_ solution, respectively.

### 3.5. Material Characterizations

The crystal structures of samples were determined by X-ray diffraction (XRD, Shimadzu XRD-6000) using high-intensity Cu Kα radiation with a wavelength of 1.54178 Å, and morphologies and element distributions by scanning electron microscopy (SEM), performed using a Hitachi S8100 machine, and energy dispersive X–ray spectroscopy (EDS). Transmission electron microscopy (TEM, Hitachi HT–7700) and high-resolution transmission electron microscopy (HRTEM, JEOL-2010 TEM) were used to determine sample microstructures. Raman spectra were obtained using a Renishaw inVia Raman spectrometer and a 532 nm laser source). Brunauer–Emmett–Teller (BET) specific surface and pore volumes of samples were measured using a Micromeritics ASAP 2460 unit. Thermogravimetric analysis (TGA) was performed on a Setaram Labsys Evo SDT Q600 at a heating rate of 10 °C min^−1^ from room temperature to 400 °C in air flow. Fourier transform infrared spectroscopy (FTIR) was carried out on an IR spectrophotometer (Shimadzu). X-ray photoelectron spectroscopy (XPS, ESCALAB 250) was used to analyze surface elemental compositions and chemical bonding.

### 3.6. Electrochemical Tests

The electrochemical properties of VN/S microrods and VN/S@PANI microrods were measured by button cell (CR2032). The mass ratio composite (70%), carbon black (20%) and polyvinylidene fluoride adhesive (PVDF, 10%) were mixed with a certain amount of (N-methylprolinodone) (NMP). The mixture is stirred into a uniform slurry and evenly coated on aluminum foil (Shanghai Aladdin Reagent Co., Ltd.) with a diameter of 14 mm and a thickness of 0.015 mm (Shanghai Aladdin Reagent Co., Ltd.). The mixture is dried in a 60 °C vacuum for 12 h. The batteries were manufactured in an Ar-filled glove box (O_2_ < 0.01 ppm, H_2_O < 0.01 ppm, Mikrouna, Super 1220/750/900). Li foil (Shanghai Aladdin Reagent Co., Ltd., Shanghai, China) is used as pair electrode/reference electrode. The electrolyte was 1 M Lithium bis(trifluoromethanesulphonyl)imide (LiTFSI, Sigma Aldrich, Shanghai, China) in a 1:1 volume ratio of 1,3-dioxolane and 1,2-Dimethoxyethane and 1 wt% LiNO_3_ additive. The Neware battery test system (Neware CT-3008) performed constant current discharge–charge tests in a potential window of 1.6 to 2.8 V. Cyclic voltammetry (CV) curves and electrochemical impedance spectroscopy (EIS) measurements (from 0.01 Hz to 100 kHz) were performed on an electrochemical workstation (ChenhuanChi-660E).

## 4. Conclusions

To sum up, a kind of porous rod-like VN with a large specific surface area was prepared by a simple calcination process. After compounded with sulfur and then coated with polyaniline, the VN/S@PANI composite showed excellent electrochemical performance. The VN/S@PANI composite delivered an initial discharge capacity of 1007 mAh g^−1^ at 0.5 A g^−1^, and it maintained at a capacity of 735 mAh g^−1^ over 150 cycles. The capacity of the VN/S@PANI cathode maintained at 410 mAh g^−1^ at 2 A g^−1^ over 400 cycles. The porous structure of VN microrods can absorb more sulfur and provide large interspace to overcome the volume change during the discharge procedure. The outer layer of PANI not only promotes the cathode conductivity but adsorbs the polysulfides and inhibits the shuttling effect of polysulfide. Thus, the VN/S@PANI cathode possessed excellent conductivity and effective chemisorption ability, which improved the electrochemical performance of Li-S batteries.

## Figures and Tables

**Figure 1 molecules-28-01823-f001:**
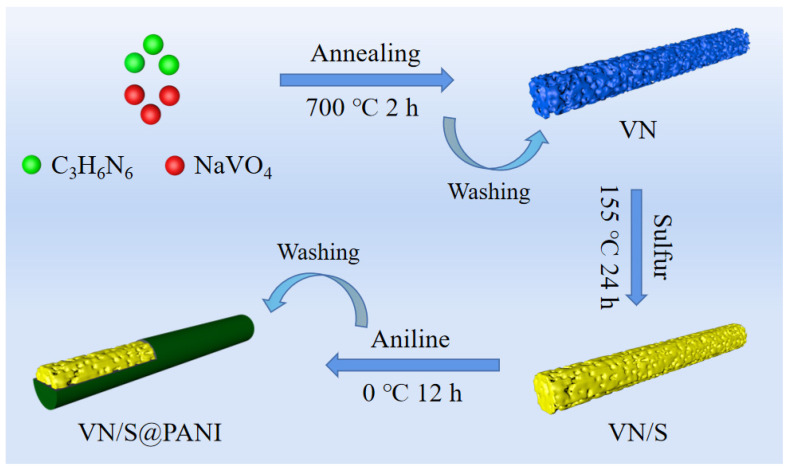
Schematic synthesis process of VN/S@PANI composite.

**Figure 2 molecules-28-01823-f002:**
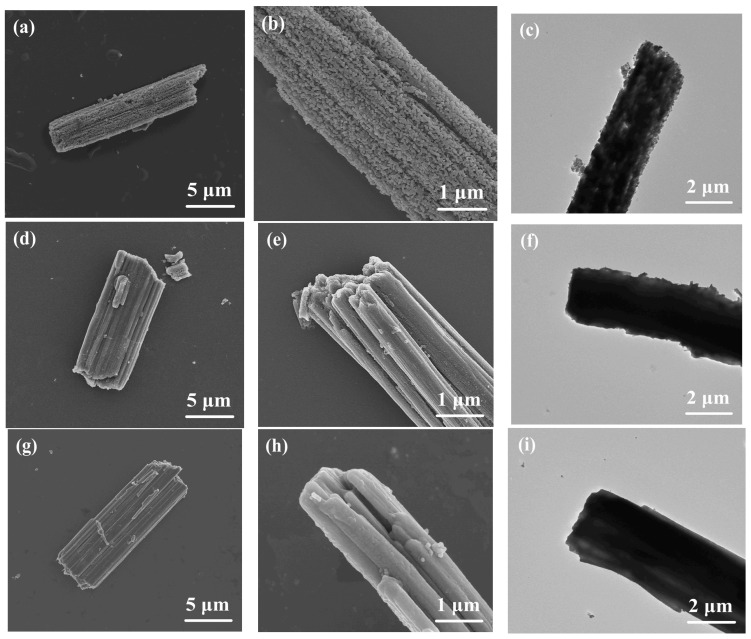
(**a**) Low-, (**b**) high-magnification SEM and (**c**) TEM images of VN. (**d**) Low-, (**e**) high-magnification SEM and (**f**) TEM images of VN/S composite. (**g**) Low-, (**h**) high-magnification SEM and (**i**) TEM images of VN/S@PANI composite.

**Figure 3 molecules-28-01823-f003:**
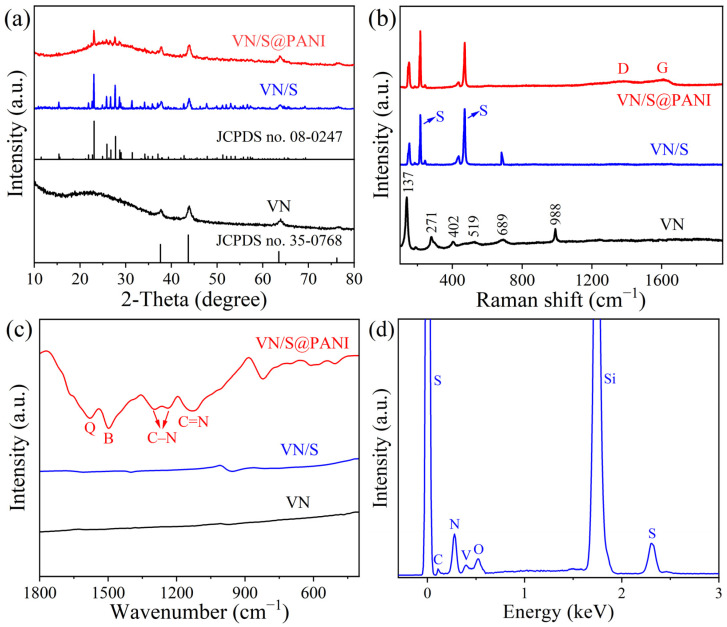
(**a**) XRD patterns, (**b**) Raman spectra and (**c**) FTIR spectra of VN microrods, VN/S microrods and VN/S@PANI microrods. (**d**) EDS analysis of VN/S@PANI microrods. (**e**) HRTEM image and (**f**) SAED pattern of VN microrods.

**Figure 4 molecules-28-01823-f004:**
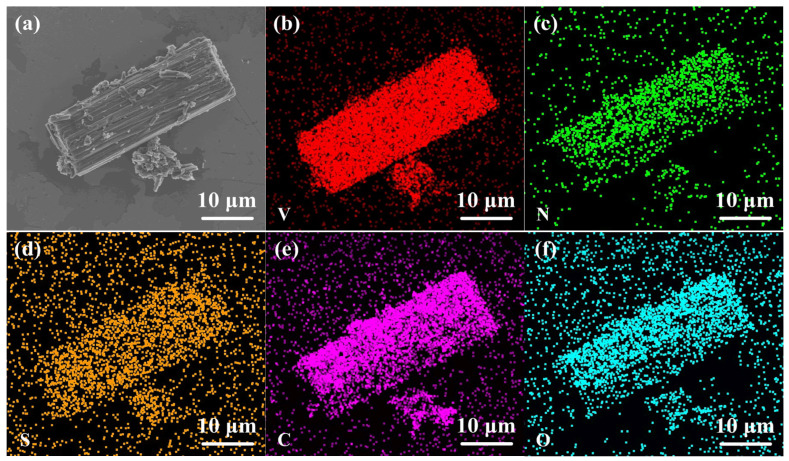
(**a**) SEM image and (**b**–**f**) elemental mapping images of VN/S@PANI composite.

**Figure 5 molecules-28-01823-f005:**
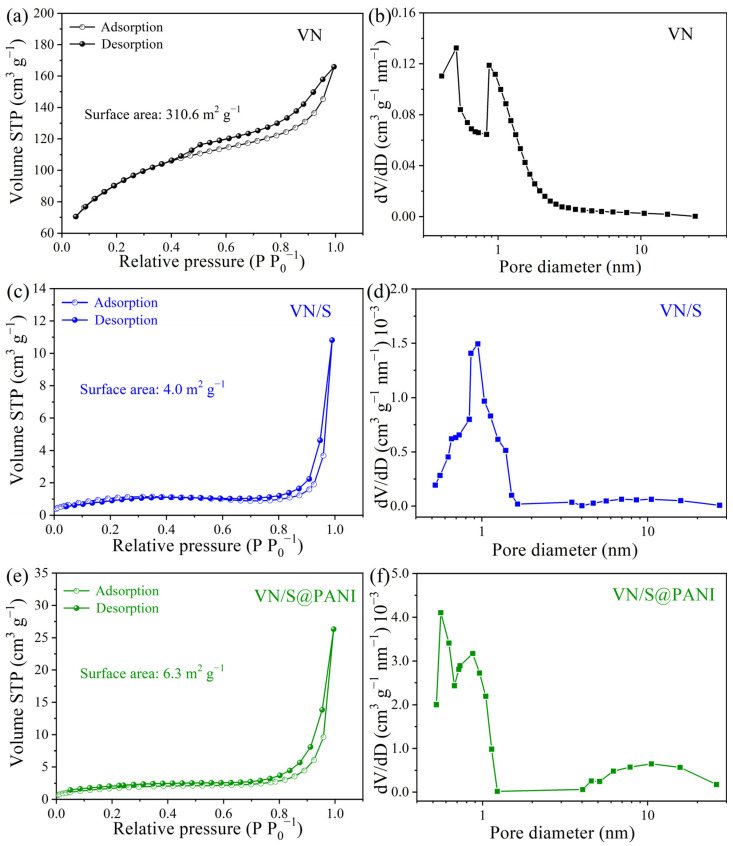
N_2_ adsorption/desorption isotherms and corresponding pore size distribution curves of (**a**,**d**) VN microrods, (**b**,**e**) VN/S microrods and (**c**,**f**) VN/S@PANI microrods.

**Figure 6 molecules-28-01823-f006:**
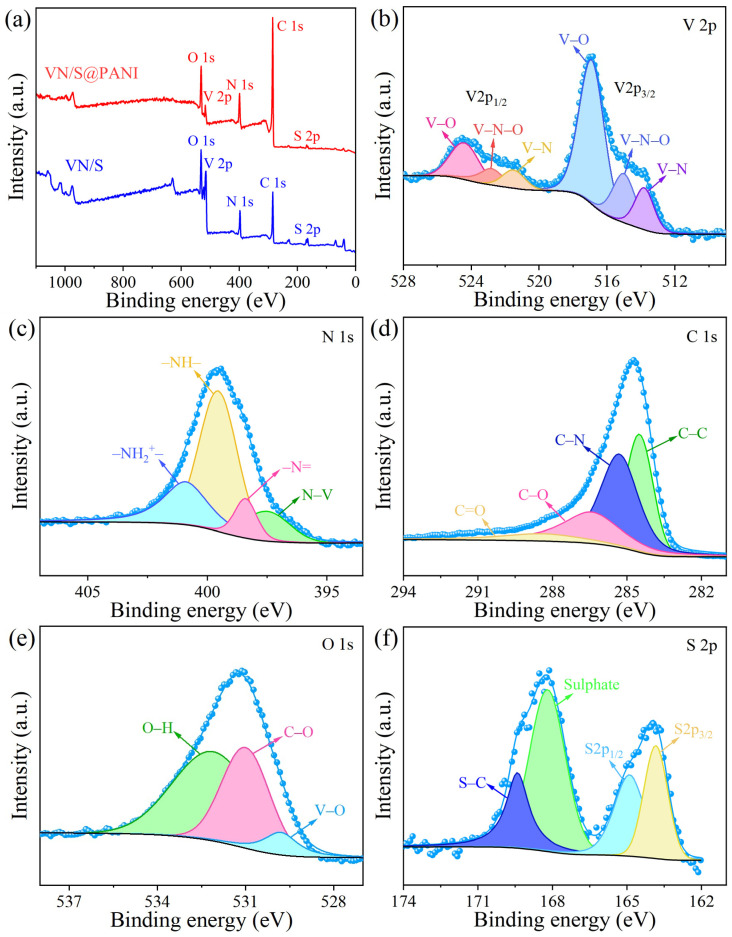
XPS spectra of VN/S@PANI microrods: (**a**) full survey spectrum, (**b**) V 2p, (**c**) N 1s, (**d**) C 1s, (**e**) O 1s, and (**f**) S 2p spectra.

**Figure 7 molecules-28-01823-f007:**
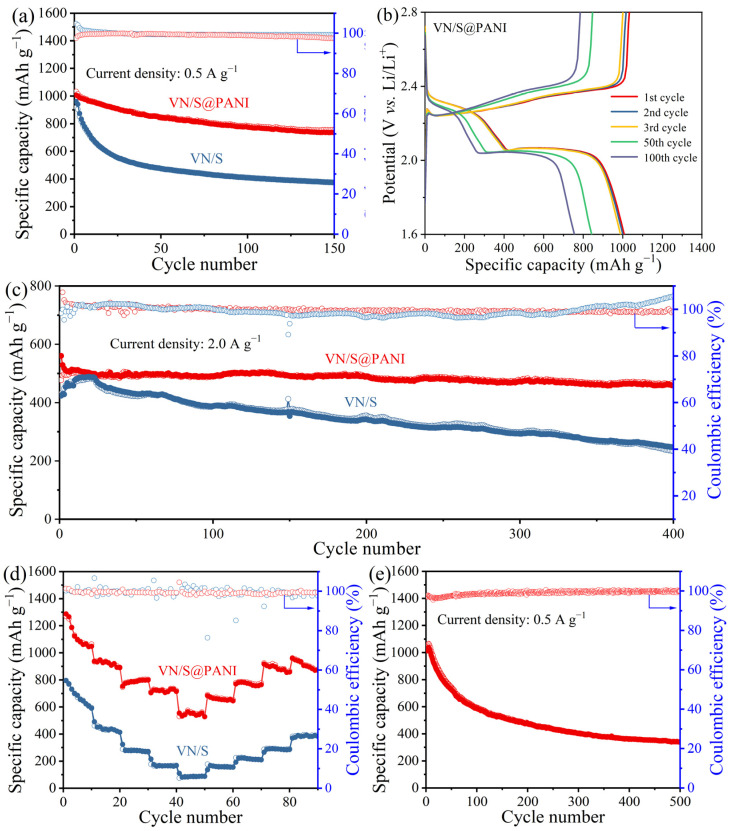
(**a**) Cycling performance of VN/S@PANI and VN/S cathodes at 0.5 A g^−1^. (**b**) Charge–discharge profiles of VN/S@PANI cathode. (**c**) Rate-performance of VN/S@PANI and VN/S cathode. (**d**) Cycling performance of VN/S@PANI and VN/S cathodes at 2 A g^−1^ over 500 cycles. (**e**) Cycling performance of VN/S@PANI cathode at 50 °C.

**Figure 8 molecules-28-01823-f008:**
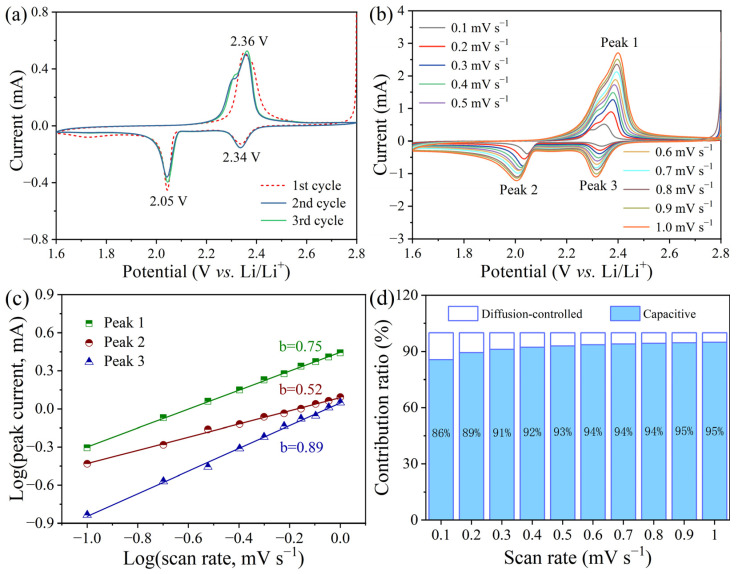
(**a**) Initial five CV curves of VN/S@PANI cathode at a scan rate of 0.1 mV s^−1^. (**b**) CV curves of VN/S@PANI cathode at 0.1 to 1.0 mV s^−1^. (**c**) The log(*i*) vs. log(*v*) of VN/S@PANI cathode. (**d**) Contribution ratio of capacitance control and diffusion control.

**Figure 9 molecules-28-01823-f009:**
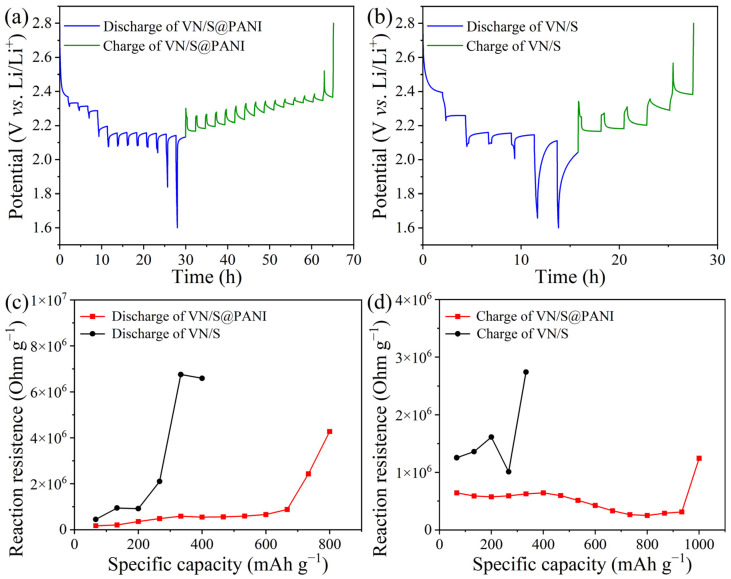
GITT time-potential distribution of (**a**) VN/S and (**b**) VN/S@PANI in discharging and charging. In situ reaction impedances of VN/S and VN/S@PANI during (**c**) discharge and (**d**) charge.

**Figure 10 molecules-28-01823-f010:**
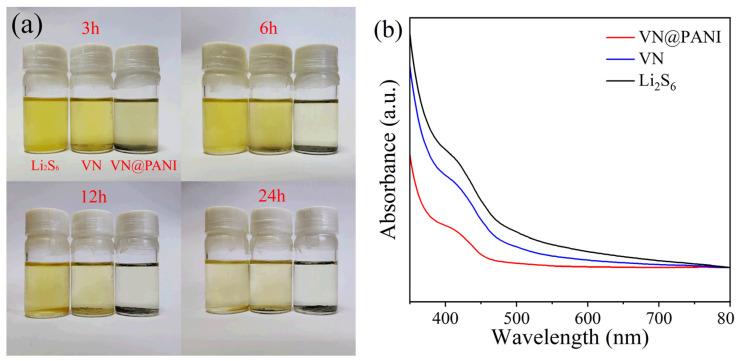
(**a**) Photograph of static adsorption test and (**b**) UV-visible spectra of Li_2_S_6_ solution, Li_2_S_6_ solution containing VN microrods and VN@PANI microrods.

**Figure 11 molecules-28-01823-f011:**
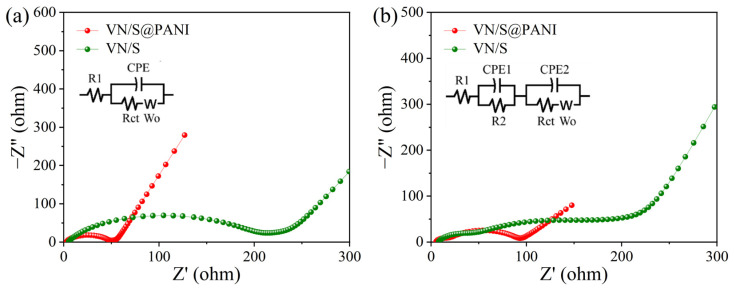
EIS plots and corresponding equivalent circuits of VN/S microrod and VN/S@PANI microrod cathodes (**a**) before the cycling test and (**b**) after 500 cycles.

## Data Availability

The data presented in this study are available on request from the corresponding author.

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
