# Peer review of "Polyaniline-Coated Porous Vanadium Nitride Microrods for Enhanced Performance of a Lithium–Sulfur Battery"

_molecules, 2023, doi:10.3390/molecules28041823_

Round 1

Reviewer 1 Report

The manuscript “Polyaniline-coated porous VN microrods for enhanced performance of lithium-sulfur battery” could be accepted after minor revision.

Comments

1.       What is VN? The author should use its complete form in the title and 1st place of the abstract and introduction.

2.       In section 3.1, “NaVO3 (0.2 g) and C3H6N6 (0.6 g) was mixed and kept in a tube furnace in N2 atmosphere and heated to 700 °C. Then, the mixtures were kept at 700 °C for 2 hours.” This statement is confusing and could be written in a single sentence.

3.       Sulfur doping is an essential process for electrochemical energy storage, and the authors should explain this in the introduction. The following papers could be helpful: https://doi.org/10.1002/cssc.202101282 and https://doi.org/10.1016/j.carbon.2022.03.043

4.       Also, PANI is a very good candidate for electrochemical energy storage, and the authors should explain this in the introduction. The following paper could be helpful: https://doi.org/10.3390/polym14020270

5.       The BET surface area of VN/S@PANI is lower than the VN, but its electrochemical performance is better than the other two. The authors could explain this in the manuscript with a piece of proper evidence and reference.

6.       The equation used to measure the capacity is missing in the manuscript.

7.       In figure 11, two different equivalent circuit diagrams are shown for a similar device. Is there any reason for this? Explain the equivalent circuit diagrams in the text.

Author Response

Dear Reviewer,

     Thank you for reconsidering our manuscript for publication, and we thank the reviewers for their helpful comments as well. We have checked and revised the manuscript carefully on the basis of the reviewers’ suggestions. All revisions to the manuscript have been marked up using the “Track Changes” function. The point-to-point responses are presented below:

Responses to Reviewer 1

Comments to the Author:

The manuscript “Polyaniline-coated porous VN microrods for enhanced performance of lithium-sulfur battery” could be accepted after minor revision.

Comment 1: What is VN? The author should use its complete form in the title and 1st place of the abstract and introduction. 

Reply: Thanks for your good comment. The full name ‘vanadium nitride’ has been added in the title, abstract and introduction parts as you suggested.

Comment 2: In section 3.1, “NaVO3 (0.2 g) and C3H6N6 (0.6 g) was mixed and kept in a tube furnace in N2 atmosphere and heated to 700 °C. Then, the mixtures were kept at 700 °C for 2 hours.” This statement is confusing and could be written in a single sentence.

Reply: The sentence has been revised as you suggested as following. NaVO3 (0.2 g) and C3H6N6 (0.6 g) was mixed and transfered into a tube furnace in N2 atmosphere and calcinated at 700 °C for 2 hours. Thanks a lot.

Comment 3: Sulfur doping is an essential process for electrochemical energy storage, and the authors should explain this in the introduction. The following papers could be helpful: https://doi.org/10.1002/cssc.202101282 and https://doi.org/10.1016/j.carbon.2022.03.043. 

Reply: Thank you very much for your good suggest. We have added the related description on page 2 in the revision as follows. For example, various S-doped carbon nanomaterials were developed and served as hosts for sulfur cathode in order to achieve rapid polysulfide redox reaction [22,23].

Comment 4: Also, PANI is a very good candidate for electrochemical energy storage, and the authors should explain this in the introduction. The following paper could be helpful: https://doi.org/10.3390/polym14020270. 

Reply: Thanks for your good suggestion. The document has been cited as ref. 24 in the revision.

Comment 5: The BET surface area of VN/S@PANI is lower than the VN, but its electrochemical performance is better than the other two. The authors could explain this in the manuscript with a piece of proper evidence and reference.

Reply: Thanks for your valuable comment. Pure VN cannot be served as the cathode material for Li-S battery. We have added the related explanation on pages 5, 8 and 14 in the revision. Page 5: Its ultra-high specific surface area with high porosity is favorable for loading more sulfur and provides more active sites for sulfur oxidation and reduction [34]. The specific surface areas of VN/S and VN/S@PANI decreased to 4.0 and 6.3 m2 g-1, respectively, demonstrating that sulfur species were composited with these sulfur hosts. Page 8: It can be seen that the polyaniline coating significantly improved its cycling stability. Page 14: Conclusions part. The outer layer of PANI not only promotes the cathode conductivity but adsorbs the polysulfides and inhibits shuttling effect of polysulfide. Thus the VN/S@PANI cathode possessed excellent conductivity and effective chemisorption ability, which improved the electrochemical performance of Li-S batteries.

Comment 6: The equation used to measure the capacity is missing in the manuscript.

Reply: We have added the equation on page 7 in the revision as follows. The specific capacity of the cathode in this work is the mass specific capacity, that is, the quantity of electricity can be discharged per unit mass of the cathode material. It can be expressed as the following formula. Sc = Id × t, where Sc is the specific capacity of the cathode, Id is the current intensity per unit mass, which is also called current density, and t is the charge and discharge time of the battery. Thanks for your good comment.

 Comment 7: In figure 11, two different equivalent circuit diagrams are shown for a similar device. Is there any reason for this? Explain the equivalent circuit diagrams in the text.

Reply: Thank you very much for your good suggestion. Figures 11a and 11b have been revised. The related explanation has been added on page 12 in the revision as follows. The inclined line corresponds to Li+ diffusion in the electrolyte, which is described as Warburg impedance (Wo) [53]. The insets in Figure 11 show the equivalent circuit diagram before and after the 500 cycle. In the equivalent circuit diagram, R1 represents the internal resistance of the electrolyte, R2 represents the internal resistance of the solid electrolyte interface (SEI) film correlated with insoluble Li2S2/Li2S, Rct is related to the charge transfer resistance and the electrode chemical kinetics. CPE1 represents the capacitance of the electrode body in the high frequency region, and CPE2 represents the capacitance of the charge transfer process at the sulfur-electrolyte interface, Wo is the semi-infinite Warburg diffusion impedance of long chain LiPSs [49,53].

Reviewer 2 Report

Manuscript ID:  molecules-2168037

 Title: Polyaniline-coated porous VN microrods for enhanced performance of lithium-sulfur battery

 Molecules

 The proposed manuscript represents a very interesting study on synthesis of polyaniline-coated VN porous microrods via a calcination, washing and polyaniline-coating process. The synthesized VN/S@PANI composite can be used in Li-S batteries. The comprehensive characterizations of obtained composite materials were presented including electrochemical studies. The used experimental methods are correct and the manuscript is logically structured. The results are well presented and the conclusions are well documented. The manuscript is valuable and I have no serious remarks/comments to its content. Some minor remarks are as follows:

  1. Page 4, line 109, the elemental scan line analysis is presented in Fig. S1 (not S2).
  2. Page 5, line 123, there is the missing superscript.
  3. The results presented in Figure 6f are not discussed in the text of the manuscript.
  4. There are missing spaces between words (e.g., line 181, 327).

 Given all mentioned above inadequacies, the manuscript can be published after minor revision.

Author Response

Dear Reviewer,

     Thank you for reconsidering our manuscript for publication, and we thank the reviewers for their helpful comments as well. We have checked and revised the manuscript carefully on the basis of the reviewers’ suggestions. All revisions to the manuscript have been marked up using the “Track Changes” function. The point-to-point responses are presented below:

Responses to Reviewer 2

Comments to the Author:

The proposed manuscript represents a very interesting study on synthesis of polyaniline-coated VN porous microrods via a calcination, washing and polyaniline-coating process. The synthesized VN/S@PANI composite can be used in Li-S batteries. The comprehensive characterizations of obtained composite materials were presented including electrochemical studies. The used experimental methods are correct and the manuscript is logically structured. The results are well presented and the conclusions are well documented. The manuscript is valuable and I have no serious remarks/comments to its content. Some minor remarks are as follows:

Comment 1: Page 4, line 109, the elemental scan line analysis is presented in Fig. S1 (not S2). 

Reply: Thank you very much for your suggestion. It has been revised as you suggested.

Comment 2: Page 5, line 123, there is the missing superscript. 

Reply: Thanks a lot. It has been revised as you suggested.

Comment 3: The results presented in Figure 6f are not discussed in the text of the manuscript. 

Reply: Thank you very much for your valuable comment. The related discussion has been added on page 7 in the revision as follows. In Figure 6f, the S 2p spectrum exhibited four peaks at 164.0, 165.0, 168.4, and 169.6 eV, which are ascribed to S-C, sulfate, S 2p1/2, and S 2p3/2, respectively. The sulfates are attributed to sulfur oxidation in the air [37].

Comment 4: There are missing spaces between words (e.g., line 181, 327). 

Reply: We have revised them as you suggested. Thank you very much for your good suggestion.